# Green Techniques for Preparation of Red Beetroot Extracts with Enhanced Biological Potential

**DOI:** 10.3390/antiox11050805

**Published:** 2022-04-20

**Authors:** Dragana Borjan, Vanja Šeregelj, Darija Cör Andrejč, Lato Pezo, Vesna Tumbas Šaponjac, Željko Knez, Jelena Vulić, Maša Knez Marevci

**Affiliations:** 1Laboratory for Separation Processes and Product Design, Faculty of Chemistry and Chemical Engineering, University of Maribor, Smetanova Ulica 17, 2000 Maribor, Slovenia; dragana.borjan@um.si (D.B.); darija.cor@um.si (D.C.A.); zeljko.knez@um.si (Ž.K.); 2Faculty of Technology Novi Sad, University of Novi Sad, Bulevar cara Lazara 1, 21000 Novi Sad, Serbia; vanjaseregelj@tf.uns.ac.rs (V.Š.); vesnat@uns.ac.rs (V.T.Š.); jvulic@uns.ac.rs (J.V.); 3Institute of General and Physical Chemistry, Studentski trg 12-16, 11000 Belgrade, Serbia; latopezo@yahoo.co.uk; 4Laboratory for Chemistry, Faculty of Medicine, University of Maribor, Taborska Ulica 8, 2000 Maribor, Slovenia

**Keywords:** red beetroot, supercritical fluid extraction, antioxidant activity, anti-inflammatory activity, antihyperglycemic activity, chemometric approach

## Abstract

Red beetroot is well known for its high proportion of betalains, with great potential as functional food ingredients due to their health-promoting properties. The objective of this study was to investigate the influence of processing techniques such as Soxhlet, cold, ultrasound and supercritical fluid extraction on the betalains content and its antioxidant, anti-inflammatory and antihyperglycemic activities. Whilst Soxhlet extraction with water has provided the highest yield, the highest content of total phenolics was found in an extract prepared using Soxhlet extraction with 50% ethanol. Amongst eight phenolic compounds detected in the extracts, protocatechuic acid was the most abundant. The concentrations of total phenolics ranged from 12.09 mg/g (ultrasound extraction with 30% methanol) to 18.60 mg/g (Soxhlet extraction with 50% ethanol). The highest anti-inflammatory activity was observed for cold extraction with 50% methanol extract. The high radical scavenging activity of supercritical fluid extracts could be a consequence of nonphenolic compounds. The chemometrics approach was further used to analyse the results to find the “greenest” method for further possible application in the processing of beetroot in the food and/or pharmaceutical industry. According to the standard score, the best extraction method was determined to be Soxhlet extraction with 50% ethanol.

## 1. Introduction

Beetroot (*Beta vulgaris* L.) is an annual or biennial crop belonging to the *Amaranthaceae* family, whose main edible part is the red tuberous root. Red beetroot has been spread worldwide, and presents one of the top ten planted vegetables associated with superior health benefits [1]. As a naturally occurring root vegetable and a rich source of phytochemicals, including betalains, polyphenols, and flavonoids, red beetroot is receiving increasing popularity for different applications in the food and pharmaceutical industries. As additives, natural betalains receive more interest than synthetic colours, due to legislative actions and growing consumer interest in the aesthetic, nutritional, and safety aspects of food and dietary supplements.

Beetroot extracts have been reported to have numerous bioactive properties, including antioxidant, anti-inflammatory, antihyperglycemic, anticancer, antihypertensive, lipid lowering effects, etc. [2,3,4,5]. It is well known that the extraction process is a crucial step in the valorisation of the plant sources; different extraction techniques and process parameters, as well as extracting solvents, can influence the final phytochemical composition and bioactive potential of obtained extracts. Recent trends in extraction techniques have largely focused on green engineering and green chemistry. Based on green principles, the main tasks of green extraction are discovering and designing extraction processes that will decrease energy consumption, allow the use of alternative solvents and renewable natural products and ensure a safe and high-quality extract/product [6].

Although there are many studies regarding beetroot, they were usually dealing with conventional extraction techniques; studies on the extraction of polyphenols and betalains by modern approaches are not readily available. The most commonly followed conventional extraction techniques are Soxhlet and cold extraction. In general, these methods refer to the extraction of bioactive compounds from plant material using different solvents, with or without heat treatment; therefore, the targeted compounds are extracted based on diffusion and mass-transfer phenomena. Conventional techniques have been well established in the industry for many years and are generally considered safe and reliable, but with some drawbacks such as time-consuming processes and high solvent consumption [7]. To overcome these mentioned drawbacks, during the last several years, significant progress has been made in extraction technology and more environmentally friendly techniques (so-called non-conventional) [8]. Ultrasound extraction represents an excellent alternative to conventional extraction, with the potential to increase the speed and efficiency of the process. This technique is based on the acoustic cavitation phenomenon, where ultrasound waves break the cellular walls, facilitate solvent penetration into plant materials, and thus enhance extraction yields [1]. Supercritical fluid extraction with CO_2_ represents another green approach that ensures selectivity in the extraction of certain target molecules by varying operating conditions such as temperature and pressure. CO_2_ is characterised by the possibility of being reused in the processing, which can reduce total energy costs in industries. In addition, co-solvents (e.g., ethanol, methanol) may be included to improve the solubility of polar compounds [9].

The aim of the present study was to characterise beetroot extracts obtained by Soxhlet, cold, ultrasound and supercritical fluid extraction in terms of their phytochemical composition and their antioxidant and pharmacological (anti-inflammatory and antihyperglycemic) activities. Phytochemical screening of obtained extracts was achieved by spectrophotometric methods, as well as by high performance liquid chromatography (HPLC). To obtain valuable information regarding the best green approach for further possible application of beetroot in the food or/and pharmaceutical industry, the optimum extraction conditions were determined by standard score analysis (using the min-max normalisation method). Principal component analysis and the artificial neural network were used to investigate the influence of active compounds on antioxidant assays.

## 2. Materials and Methods

### 2.1. Plant Material

Lyophilised beetroot material was obtained from Alfred Galke GmbH (Bad Grund, Germany).

### 2.2. Chemicals

Methanol (CAS Reg. No. 67-56-1) and ethanol (CAS Reg. No. 64-17-5) with purity ≥ 99.9%, used as solvents for conventional extractions, were purchased from Sigma-Aldrich Chemie GmbH (Steinheim, Germany). Carbon dioxide (CAS Reg. No. 124-38-9) was purchased from MESSER (MG-Ruše, Slovenia), with a purity of 99.99%. All standards, reagents and chemicals used in the study were of analytical grade, purchased from Sigma Chemicals Co., Merck (St. Louis, MO, USA), J.T.Baker (Deventer, The Netherlands) and Lachner (Brno, Czech Republic).

### 2.3. Preparation of Beetroot Extracts

Beetroot material was ground, and the average particle diameter of the material, subjected to further extraction was determined (0.5 mm). Beetroot samples were extracted using different extraction methods—ultrasound, Soxhlet, cold and supercritical fluid. Material from the same batch has been applied in all experiments. Furthermore, various solvents and co-solvents were employed. Afterwards, the obtained extracts were evaporated (BÜCHI Rotavapor R-114 and BÜCHI Vacuum Controller B-721, Uster, Switzerland), and the solvent was removed to dry under reduced pressure. All obtained extracts were stored at −20 °C until further assays.

Extraction yield (mass of extract/mass of dry material) was used to indicate the effects of the extraction conditions.

#### 2.3.1. Ultrasound Extraction (UE)

Dried and ground material (20 g) was introduced to an Erlenmeyer flask, and 250 mL of solvent was added, where different mixtures were used as the solvent—30% aqueous methanol (UE 30% MeOH), 50% aqueous methanol (UE 50% MeOH), 50% aqueous ethanol (UE 50% EtOH) and water (UE H_2_O). Then the Erlenmeyer flask was immersed into an ultrasonic bath (Iskra-Pio, Slovenia) at a fixed power of 40 kHz, with the liquid level in the Erlenmeyer flask kept lower than that of the bath. Extraction was performed at a constant temperature of 40 °C for 1.5 h.

#### 2.3.2. Soxhlet Extraction (SE)

The Soxhlet extraction was performed using the Soxhlet apparatus ISOLAB NS29-32 (Merck KGaA, Darmstadt, Germany). Different solutions were employed as the solvent—30% aqueous methanol (SE 30% MeOH), 50% aqueous methanol (SE 50% MeOH), 50% aqueous ethanol (SE 50% EtOH) and water (SE H_2_O). A total of 20 g of dried and ground material was introduced into the tube, and 150 mL of solvent was added to the flask. Extraction was carried out in three cycles for approximately 2 h. The heating temperature was adjusted to the boiling point of the employed solvent.

#### 2.3.3. Cold Extraction (CE)

Dried and ground material (20 g) and solvent (250 mL) were added into an Erlenmeyer flask. Various solutions were employed as the solvent—(CE 50% MeOH), 50% aqueous methanol (CE 80% MeOH), 50% aqueous ethanol (CE 50% EtOH) and water (CE H_2_O). To avoid constant stirring, magnetic grain was added to the mixture, and it was then placed on a magnetic stirrer. The extraction took place for about 2 h at room temperature.

#### 2.3.4. Supercritical Fluid Extraction (SFE)

Beetroot was extracted using supercritical carbon dioxide and two different co-solvents (ethanol and propanol). Experiments were performed on a semi-continuous high-pressure flow apparatus designed for a maximum pressure of 500 bar and a temperature of 100 °C. The procedure of the lab-scale extraction process has been described in previous research [10]. Extractions were carried out in cycles at pressures of 100 bar and 300 bar and temperatures of 40 °C and 60 °C (SFE EtOH 40 °C 100 and 300 bar; SFE EtOH 60 °C 100 and 300 bar; SFE PrOH 40 °C 100 and 300 bar; SFE PrOH 60 °C 100 and 300 bar). Approximately 15 g of dried ground material was charged into the extractor (V = 60 mL). The temperature of the water bath was controlled and kept constant (±0.5 °C, LAUDA DR. R Wobser GmbH & Co. KG, Lauda Königshofen, Germany). The apparatus was first cleaned with nitrogen, and then the gas was employed in the extraction process. After that, liquefied gas (CO_2_) was continuously introduced using a high-pressure pump (ISCO syringe pump, model 260D, Lincoln, NE, USA, Pmax = 450 bar) through the preheating coil and over the bed of the sample in an extractor. The solvent flow was determined with a flow meter (ELSTER HANDEL GmbH, Mainz, Germany). The product was separated in a separator (glass trap), where the precipitation was performed under atmospheric conditions.

### 2.4. Phytochemical Analysis

#### 2.4.1. Phenolics Profiling (TPh)

Total phenolic content (TPh) in the extracts was established using the Folin–Ciocalteau spectrophotometric method adapted to microscale [11]. The sample (15 μL) was mixed with distilled water (170 μL), 2 N Folin–Ciocalteu’s reagent (12 μL) and 20% Na_2_CO_3_ (30 μL) in a plate well. Absorbance was measured after 1 h (room temperature, dark conditions) at 750 nm, using distilled water as blank. Gallic acid (GAE) was used for the calibration curve.

Chromatographic analysis for identification and quantification of phenolic compounds were carried out as recommended by Tumbas Šaponjac et al. [11], using Shimadzu Prominence HPLC, connected to an SPD-20AV UV/VIS detector (Shimadzu, Kyoto, Japan), with Luna C-18 RP column, 5 lm, 250 mm × 4.6 mm with a C-18 guard column, 4 mm × 30 mm (Phenomenex, Torrance, CA, USA). Gradient elution was applied using acetonitrile (A) and water acidified with 1% formic acid in d-water (B), at flow rates of 1 mL/min, at the following order: 10% to 25% A (0–10 min); 25% to 60% A (10–20 min); 60% to 70% A (20–30 min); 70% to 10% A (30–40 min); 10% A (5 min) (equilibration time). For hydroxybenzoic acids, chromatograms were recorded at 280 nm; for hydroxycinnamic acids at 320 nm; and for flavonoids at 360 nm. HPLC standards were dissolved in 50% methanol.

#### 2.4.2. Betalains Profiling (TBc and TBx)

The method of Von Elbe et al. [12], adapted to microscale, was used for total betalain estimation. The sample was diluted with 0.05 M phosphate buffer pH 6.5 in a plate well, up to the final volume of 250 μL, using the same buffer as blank. For total betacyanin (TBc) and betaxanthin (TBx) content, absorbances were read immediately at 538 and 476 nm, respectively, while the correction was estimated at 600 nm. Absorbances were further calculated using the Equations (1)–(3):X = 1.095 × (a − c),(1)
Y = b − Z − X/3.1,(2)
Z = a − X,(3)
a—absorbance at 538 nm, b—absorbance at 476 nm, c—absorbance at 600 nm, X—absorbance of betanin corrected for coloured impurities, Y—absorbance of vulgaxanthin-I corrected for coloured impurities and Z—absorbance of impurities. Red and yellow pigment concentration (C) in the extracts was calculated with the Equation (4):C (mg/100 mL) = X(Y) × F × 1000/A1%,(4)
where F is the dilution factor (25 or 12.5) and A1% is the absorbance coefficient (1120 for betanin, 750 for vulgaxanthin). The total content of betacyanin in the extracts was expressed as mg betanin equivalents (BE) per 100 g of plant material. The total content of betaxanthin in extracts was expressed as mg vulgaxanthin (VE) per 100 g of plant material.

### 2.5. In Vitro Antioxidant Analysis

#### 2.5.1. DPPH free Radical Scavenging Assay (DPPH)

The DPPH radical scavenging assay was performed spectrophotometrically according to the method of Tumbas Šaponjac et al. [11]. Briefly, 250 μL DPPH• solution in methanol (0.89 mM) was mixed with 10 μL of extract in a microplate well. Absorbance was measured at 515 nm after 50 min incubation in the dark at ambient temperature. Methanol was used as the blank. The DPPH radical scavenging activity values were calculated using the Equation (5):DPPH = [(Acontrol − Asample)/Acontrol] × 100,(5)
where Acontrol is the absorbance of the blank and Asample is the absorbance of the extract sample. The results were expressed in μmol Trolox equivalent (TE) per 100 g of plant material.

#### 2.5.2. ABTS Radical Scavenging Assay (ABTS)

The ABTS radical scavenging assay was evaluated employing a modified method according to Tumbas Šaponjac et al. [13]. The absorbances of 250 μL activated ABTS+● (with MnO_2_), before and 35 min (incubated at 25 °C) after the addition of 2 μL of extract, were measured at 414 nm. Water was used as the blank. The results were expressed as μmol Trolox equivalent (TE) per 100 g of plant material.

#### 2.5.3. Reducing Power (RP)

Reducing power was analyzed according to Oyaizu’s method [14], additionally adapted for a microplate. The sample/water (25 μL), sodium phosphate buffer (Ph = 6.6) (25 μL), and 1% K_3_[Fe(CN)_6_] (25 μL) were mixed, incubated (20 min at 50 °C), cooled and mixed with 10% CCl_3_COOH (25 μL), and centrifuged (2470× *g* for 10 min). Supernatant (50 μL) was mixed with distilled water (50 μL) and 0.1% FeCl_3_ (10 μL) in the plate well, and absorbances at 700 nm were measured immediately. Trolox was used as the calibration standard (Trolox equivalent (TE) per 100 g of plant material).

#### 2.5.4. β-Carotene Bleaching Assay (BCB)

The β-carotene bleaching capacity of the samples was evaluated by the β-carotene linoleate model system of Al-Saikhan et al. [15]. The absorbances of all the samples were taken at 470 nm at zero time and after 180 min, while during this time, the microplate was incubated at 50 °C. The results were expressed as μmol Trolox equivalent (TE) per 100 g of plant material.

### 2.6. In Vitro Pharmacological Analysis

#### 2.6.1. Anti-Inflammatory Activity Assay (AIA)

The anti-inflammatory activity was determined by protein denaturation bioassay using egg albumin (from fresh hen’s eggs), according to the method adopted by Ullah et al. [16]. Briefly, 2 mL of extract was incubated with 0.2 mL of egg albumin and buffered saline phosphate (pH 6.4) at 37 °C for 15 min and then at 70 °C for 5 min. After cooling, the absorbance was measured at 660 nm.

#### 2.6.2. Antihyperglycemic Activity Assay (AHgA)

To examine in vitro antihyperglycemic activity, α-glucosidase inhibitory potential was performed, using the method reported by Tumbas Šaponjac et al. [13]. L−14-nitrophenyl α-D-glucopyranoside (2 mmol, 100 μL) and samples (c = 250 mg/mL, 20 μL), both dissolved in 10 mmol/L potassium phosphate buffer (pH 7.0), were mixed in a plate well with the enzyme solution (56.66 mU/mL, 100 μL) to initiate the reaction. After incubation at 37 °C for 10 min, the absorbance was measured at 405 nm.

### 2.7. Chemometric Approach

#### 2.7.1. Standard Scores (SS)

The ranking between extracts was performed, based upon the ratio of raw data and extreme values for each applied assay [17] according to Equation (6):(6)$${\bar{x}}_{i}=\frac{x_{i}-\min_{i}x_{i}}{\max_{i}x_{i}-\min_{i}x_{i}},\;\forall i$$

#### 2.7.2. ANN Modelling

A multi-layer perceptron model (MLP), with three layers (input, hidden and output) was employed for the ANN models construction. This ANN model is well known and widely accepted for its high capability of approximating nonlinear functions [18,19,20]. Before the calculation, input and output data was normalised (min-max normalisation) to improve the calculation of the ANN [21]. Throughout the process of computation and variation of the ANN structure, input values are permanently transferred to the network inputs [21,22]. The learning cycle of the ANN construction process was replicated 100,000 times, screening the various topologies of ANN models, introducing a distinct number of neurons in the hidden layer (between 5 and 20), and several activation functions (for instance: logarithmic, logistic, tangent hyperbolic, or identity), including arbitrarily chosen initial values of weight coefficients and biases. The optimisation of the ANN structure was accomplished by limiting the validation error. BFGS calculation was employed for investigating the solution of the unconstrained nonlinear optimisation during the ANN modelling [21]. ANN was developed to predict antioxidant and pharmacological in vitro assays such as: DPPH, ABTS, RP, BCB, AIA and AHgA, according to the concentration of polyphenol compounds determined by HPLC analysis.

Coefficients associated with the hidden and output layer (weights and biases) are grouped in matrices *W*_1_ and *B*_1_, and *W*_2_ and *B*_2_, respectively [18]:(7)$$Y=f_{1}(W_{2}\cdot f_{2}(W_{1}\cdot X+B_{1})+B_{2})$$

#### 2.7.3. Global Sensitivity Analysis

The global sensitivity calculation was performed, according to the weight coefficients obtained during the development of the ANN model, employing Yoon’s equation [23]. The relative influence of the input variables on the output variable was calculated according to:(8)$$RI_{ij}(\%)=\frac{\sum_{k=0}^{n}\left(w_{ik}\cdot w_{kj}\right)}{\sum_{i=0}^{m}\left|\sum_{k=0}^{n}\left(w_{ik}\cdot w_{kj}\right)\right|}\cdot 100\%$$
where: *RI*—the relative impact of input variables on specific output, *w*—weight coefficient in the ANN model (defined by Equation (7)), *i*—input of the ANN model, *j*—output of the ANN model, *k*—hidden neuron of the ANN model, *n*—number of neurons in the hidden layer and *m*—number of inputs in the ANN model.

### 2.8. Statistical Analyses

The data were processed statistically using the software package STATISTICA 10.0 (StatSoft Inc., Tulsa, OK, USA). All determinations were made in triplicate, all data were averaged, and they were expressed by mean values. The principal component analysis (PCA) was used to discover possible correlations among measured parameters and to classify objects into groups.

## 3. Results and Discussion

### 3.1. Extraction Yield

The choice of a method for isolating active components with the highest yield and purity from natural sources is mostly dependent on the nature of the compounds and raw material that will be processed. There are many conventional and non-conventional methods used to recuperate antioxidants from plants. Nevertheless, extraction yield and antioxidant activity not only vary by the extraction method used, but also by the solvent employed for extraction. A suitable solvent and working condition should be chosen to extract aimed antioxidants using several extraction methods, as different solvents would yield different extracts and extract compositions [24]. Polar solvents are commonly used for recovering polyphenols from plant matrices. The most suitable solvents are aqueous mixtures containing acetone, ethanol, ethyl acetate and methanol. Ethanol has been recognised as a suitable solvent for polyphenol extraction and is not harmful for human consumption. Methanol is generally more effective in extracting lower molecular weight polyphenols, whereas aqueous acetone is suitable for extracting higher molecular weight flavanols [25]. Optimisation of the process variables is also essential in order to consider that the aim of the extraction process is not the highest extraction yield, but the lowest use of money and energy.

In the frame of this study, two conventional (SE and CE) and two non-conventional (UE and SFE) techniques were applied, being conscious of the principles of green chemistry. The extraction with water resulted in the highest extraction yield, especially for ultrasound and Soxhlet extraction, whereas the cold extraction gave a bit lower yield. The extraction yields with different solvents presented in the following order: water > 50% ethanol > 50% methanol > 30% methanol, as shown in Figure 1.

For the effectiveness of extracting method, the results showed that yields of the extract were better when extraction was done under reflux (SE). This indicates that hot solvent systems under reflux state are more efficient for recovering antioxidant components, thus offering higher extract yields. As shown in Figure 1a, the extraction yield with different methods presented in the following order: SE > UE > CE.

On the other hand, the yield of SFE with carbon dioxide and both co-solvents, ethanol and isopropanol, was significantly lower. The yield obtained under different conditions of pressure and temperature is shown in Figure 1b; the highest value was obtained with isopropanol as a co-solvent under the pressure of 300 bar and a temperature of 60 °C, but this value was still considerably lower than that obtained using the above-mentioned extraction method. This can be explained by the difference in chemical characteristics and polarities of various antioxidant compounds and the particular solvent [26].

### 3.2. Phytochemical Analysis

The chemical composition of obtained extracts in terms of TPh, TBc and TBx were tested using spectrophotometric assays (Table 1); presented results showed significantly different content of target compounds among samples, depending on the applied extraction method.

Achieved total phenolic content in beetroot extracts ranged from 76.05 mg GAE/g to 226.80 mg GAE/g. The highest content of total phenolics was found in an extract prepared using SE 50% EtOH, but no significant differences were observed when compared to UE 30% EtOH and SE H_2_O. When it comes to environmentally friendly approaches, SE is one of the conventional techniques. However, there are some drawbacks to SE, including a large amount of solvent usage, long extraction time and excessive loss of heat energy. On the other hand, UE is an innovative “clean” technique that has gained popularity due to its excellent advantages compared to conventional methods, such as the small quantities of solvent required, short extraction time and low economic and environmental impacts [27]. In SFE samples, phenolic compounds were not determined. Precisely, carbon dioxide is a non-polar molecule, while targeted compounds (e.g., polyphenols and betalains) are mostly polar.

The available literature data indicate that the phenolic content in water extracts of beetroot cultivated under different conditions ranged from 167 mg/100 g to 537 mg/100 g [28]. In the study of Yasaminshirazi et al. [29], the phenolic compounds of 15 organic beetroot genotypes were analysed; extraction was performed using CE and methanol as a solvent. The total phenolics in red-coloured genotypes ranged from 352.46 mg GAE/100 g to 489.06 mg GAE/100 g. On the other hand, Roboczki et al. reported the significantly lower total phenolic content in beetroot genotypes grown in Hungary, which varied between 45.47 and 83.37 mg GAE/100 g. Kovarovič et al. [30] investigated the total phenolics in four beetroot varieties in the Czech Republic, extracted by CE and 80% ethanol. The authors noted the range from 36.87 mg/100 g to 88.77 mg/100 g, which is significantly lower than the values measured in this study when 50% EtOH was used as extraction solvent. Many authors reported that apart from the efficiency of the extraction method, the total content of the phytochemicals may be affected by the growing area, agrochemical composition and other environmental factors involved in the production of bioactive constituents. In a recent study by Lazar et al. [31], the optimisation of bioactive compound extraction from beetroot peel was investigated; the interaction of time (49.9 min), temperature (52.52 °C), citric acid (1.5%) and ethanol concentration (50%) led to the improvement of bioactives extraction in terms of the highest betalains and phenolics of 239 mg GAE/100 g.

Betalains are water-soluble natural pigments, which include red-violet betacyanins and yellow-orange betaxanthins. The intensity of beetroot colour depends on the ratio between betacyanins and betaxanthins, and the literature reveals that betacyanin compounds make up more than 80% of all beet pigments [32]. Regarding the TBc in the present study, the content in the examined extracts ranged from 2.52 mg BE/100 g to 4.17 mg BE/100 g (Table 1). Conventional extraction techniques, i.e., SE and CE, exhibited the highest extraction efficiency of these pigments. The results of this study indicated a nearly similar range for the TBx values; they ranged from 2.49 mg VE/100 g to 3.77 mg VE/100 g.

Recent studies also investigated approaches such as maceration and Soxhlet extraction to recover betalains from red beetroot. Delgado-Vargas et al. [33] and Wiczkowski et al. [34] reported that the addition of ethanol or methanol to water is generally necessary to thoroughly extract the pigments in the presence or absence of acidification or heat treatment, which is also confirmed in this study. In terms of non-conventional techniques, to this day, several approaches have been used for the extraction of betalains. Ultrasound-assisted extraction [35], microwave-assisted extraction [36], membrane processing [37] and gamma irradiation [38] are the processes that have advantages over conventional procedures as alternative environmentally friendly procedures for betalains extraction, but they also possess several drawbacks (purity of extracts, expensive equipment and procedures, etc.). Hence, irrespective of the technique used for extraction, it is recommended to optimise the processing conditions to accelerate the recovery of betalains for different purposes.

For the deeper analysis of polyphenols, HPLC analysis was conducted, and the obtained results are presented in Table 2. In all beetroot extracts, the presence of eight compounds was detected, and their concentrations were in the range of 12.09 to 18.60 mg/g. SE 50% EtOH and UE 30% EtOH exhibited the highest concentrations of polyphenols, which is in accordance with results obtained by spectrophotometric assay. Observed disagreement in the values relative to spectrophotometric analysis can be explained by the fact that the Folin–Ciocalteu assay is non-selective and not an absolute measure of the amount of phenolic compounds [39]. In addition, components such as proteins, amino acids, carbohydrates, unsaturated fatty acids, vitamins, aldehydes and ketones were classified as possible contributors to the final results of the Folin–Ciocalteu assay [40].

It may be noticed that protocatechuic acid was the most abundant phenolic acid in all beetroot extracts, followed by gallic acid, epicatechin, vanillic acid, gentisic acid, chlorogenic acid, caffeic acid and coumaric acid. Vulić et al. [41] reported the presence of protocatechuic, caffeic and ferulic acids in four beetroot cultivars; the contents of phenolics in beetroot pomaces varied from 1.87 mg/g to 11.98 mg/g. In the study of Tumbas Šaponjac et al. [11], the most abundant compounds were catechin and protocatechuic acid.

### 3.3. In Vitro Antioxidant and Pharmacological Activities

In practice, for the in vitro assessment of antioxidant activity of endogenous phytochemicals, a single assay is not sufficient; different assays vary in terms of mechanisms and experimental conditions. In addition, antioxidant molecules differ in polarities, thus they can act as radical scavengers by the electron-donating mechanism or by the hydrogen-donating mechanism. The antioxidant activity of beetroot extracts was challenged using four methods; all samples showed different activities in relation to the applied method. 

According to the results presented in Table 3, SFE extracts exhibited the highest antioxidant activity for all applied assays. Despite the fact that in SFE polyphenols and betalains were not detected, the high antioxidant activity of these extracts could be a consequence of the presence of other nonphenolic compounds. SFE EtOH 60 °C 300 bar extract exhibited the best results in three (RC, ABTS and BCB) out of four investigated tests, while SFE PrOH 40 °C 300 bar showed the best result in the DPPH test. Thus, we used the same type of extraction, but different under conditions.

Betalains above temperatures of 50 °C could thermally degrade. The thermal degradation of betacyanins, mainly from red beet, has been studied widely, and it was found that this causes the degradation of betacyanins to betalamic acid and cyclodopa-5-Oglucoside [42,43,44,45]. Betanin in alkaline conditions could also degrade to yellow betalamic acid and colorless cyclodopa-5-O-glucoside [42,44,45]. For example, in the study by Gandía-Herrero et al. (2012), the capacity for scavenging ABTS•+ from betalamic acid was better when compared to that of trolox [46]. The same compound was also able to reduce Fe^3+^ to Fe^2+^. These activities were attributed to the extended conjugated system of betalamic acid, nevertheless, it was dependent on the pH of the environment: pH values above 5.5 increased the activity of betalamic acid. Other thermal degradation pathways are various decarboxylation reactions and the removal of the glycoside unit [42,43,44,45].

Two spectrophotometric tests were applied to investigate the biological activities of beetroot extracts. Two beetroot extracts exhibited the best anti-inflammatory activity: CE 50% MeOH (42.86%) and CE 50% EtOH (42.68%) at the concentration 10 mg/mL. In this study, diclofenac sodium was used as a standard anti-inflammatory drug, and at a concentration of 0.25 mg/mL, exhibited 47.65%. However, the anti-inflammatory effect could be related to the free radical scavenging activity, and the effect depends on a synergic action of all the components [47].

Protein denaturation is a process where due, to external factors such as heat, strong acid or strong base, an organic solvent or a concentrated inorganic salt causes the disorientation of the protein’s tertiary and secondary structure. Nonsteroidal anti-inflammatory drugs (NSAIDs) are commonly prescribed medications due to their verified effectiveness in reducing pain and inflammation. The denaturation of protein causes the production of autoantigens in conditions such as rheumatic arthritis, cancer and diabetes, which are conditions of inflammation. Hence, by the inhibition of protein denaturation, inflammatory activity can be inhibited [48].

SFE EtOH 40 °C 300 bar showed the best AHgA. At the concentration of 10 mg/mL, it showed 88.08%; the positive control, acarbose, showed 67.72% AhgA at a concentration of 0.0025 mg/mL. The same extract showed the best result in the spectrophotometric DPPH test.

Antioxidant activity can be expressed through various mechanisms. SFE extracts contain other phytochemicals that have lower polarity, due to the fact that supercritical CO_2_ in the presence of a polar cosolvent has been used for extraction. Beetroot is rich in carbohydrates, fat, protein, micronutrients and several functional constituents exhibiting substantial health-promoting properties. Among the less polar and non-polar compounds; beetroot contains a considerable amount of carotenoids and both essential and non-essential amino acids such as tryptophan, isoleucine, leucine, lysine, threonine, methionine, phenylalanine, tyrosine, valine, cystine, arginine, histidine, alanine, glutamic acid, glycine, proline, aspartic acid and serine [49]. These compounds act as anticarcinogens and immunoenhancers, involve pro-vitamin A activity and possess antioxidant ability. Supercritical fluid extraction was used to extract free amino acids from sugar beet and sugar cane molasses [50], indicating the presence of these compounds in supercritical extracts.

### 3.4. Chemometrics Approach

The standard score (SS) is obtained by summing the normal scores for nine variables (TPh, TBc, TBx, DPPH, ABTS, RP, BCB, AIA and AHgA), which was then multiplied by its weight:(9)$$\mathrm{SS}=w_{1}\cdot \bar{\mathrm{TPh}}+w_{2}\cdot \bar{\mathrm{TBc}}+w_{3}\cdot \bar{\mathrm{TBx}}+w_{4}\cdot \bar{\mathrm{DPPH}}+w_{5}\cdot \bar{\mathrm{ABTS}}+w_{6}\cdot \bar{\mathrm{RP}}+w_{7}\cdot \bar{\mathrm{BCB}}+w_{8}\cdot \bar{\mathrm{AIA}}+w_{9}\cdot \bar{\mathrm{AHgA}}$$

If the value of the SS function is close to 1, it shows the tendency of the tested processing parameters of being optimal. The maximum of function SS represents the optimal extraction parameters. The standard scores are presented in Figure 2.

According to the standard score, the best extraction method was SE 50% EtOH, with an SS value 0.674, while a very good extraction result was gained with the SE H_2_O extraction procedure, with an obtained SS value of 0.629. Good extraction results were obtained using UE 50% EtOH, SE 30% MeOH and CE 50% EtOH methods, with standard score values of 0.568, 0.581 and 0.561, respectively.

The PCA of the presented data explained that the first three principal components explained 68.61% of the total variance (33.50; 22.57 and 12.53%, respectively) in the seventeen variables space (Figure 3). Considering the results of the PCA of the chemical parameters, *e*Cat (epicatechin) (which contributed 11.3% of the total variance), AHgA (12.8%), Caff (caffeic acid) (14.1%), *p*Cat (protocatechuic acid) (15.4%) and Gal (gallic acid) (16.4%) exhibited positive scores according to the fist principal component PC1. A positive contribution to the PC2 calculation was observed for TPh (7.5% of the total variance), ABTS (7.1%), Chl (chlorogenic acid) (9.5%), RP (17.4%), BCB (18.9%) and Cum (coumaric acid) (13.2%), while a negative score for PC2 calculation was observed for TBx content (18.1%). A positive contribution to the PC3 calculation was observed for Van (vanillic acid) content (39.4% of the total variance), whereas a negative score for PC3 calculation was observed for the AIA value (34.4%).

According to the ANN performance, the optimal number of neurons in the hidden layer was 8 (network MLP 11-8-6) to obtain high values of R^2^ (overall 0.999) and low values of the sum of squares (SOS). The applied training algorithm was BFGS 197, with Tanh for hidden activation and Identity for output activation function. ANN models were used to predict antioxidant and pharmacological assays (DPPH, ABTS, RP, BCB, AIA and AHgA) reasonably well for a broad range of input values (concentration of individual phenolic compounds: Gal, *p*Cat, *e*Cat, Van, Chl, Cum, Gen and Caff, and also TPh, TBc and TBx).

The ANN predicted values were very close to the target values of the antioxidant assays, in terms of R^2^ value, for the ANN models. The SOS obtained with ANN models are of the same order of magnitude as the experimental errors reported in the literature [18]. The ANN model is complex (150 weights-biases) because of the high non-linearity of the developed system [21,22].

Table A1 (Appendix A) presents the elements of matrix *W*_1_ and vector *B*_1_ (presented in the bias column), and Table A2 (Appendix A) presents the elements of matrix *W*_2_ and vector *B*_2_ (bias) for the hidden layer, used for ANN calculation (Equation (7)). The obtained ANN model demonstrated an adequate generalisation capability for antioxidant and pharmacological assays data prediction.

The influence of input variables, i.e., concentration of Gal, *p*Cat, *e*Cat, Van, Chl, Cum, Gen and Caff and TPh, TBc and TBx on the antioxidant assays (DPPH, ABTS, RP, BCB, AIA and AHgA) were studied, based on Yoon’s interpretation method of a developed ANN model. The graphical presentation of the Yoon’s analysis for the ANN model results is shown in Figure 4.

According to Figure 4, Van, Chl, Cum and TBx were the most influential parameters on RP, with an approximate relative importance of +14.14, +9.07, +10.01 and +8.89%, respectively, while the influence of Gen and TBc were negative, -18.03 and −15.79%, respectively. 

*e*Cat, Van, Chl and TBx were the most influential parameters on ABTS, with an approximate relative importance of +7.48, +17.01, +11.82 and +10.01%, respectively, while the influence of *p*Cat and TPc were negative, –9.79 and −14.57%, respectively.

Van, Chl and Cum were the most influential parameters on BCB, with an approximate relative importance of +11.63, +8.92 and +16.41%, respectively, while the influence of *p*Cat and TPc were negative, −15.88 and −13.49%, respectively.

*e*Cat, Chl and TBx were the most influential parameters on AIA, with an approximate relative importance of +12.04, +15.91 and +13.11%, respectively, while the influence of Van and TPc were negative, −21.78 and −13.57%, respectively.

*p*Cat, Caff and TPh were the most influential parameters on AHgA, with an approximate relative importance of +17.79, +14.95 and +12.22%, respectively, while the influence of Chl and Cum were negative, −14.17 and −9.86%, respectively.

*p*Cat, Van and Caff were the most influential parameters on DPPH, with an approximate relative importance of +11.30, +17.37 and +11.12%, respectively, while the influence of *e*Cat and Gen were negative, −12.37 and −14.89%, respectively.

## 4. Conclusions

This study aimed to investigate the influence of isolation techniques such as Soxhlet, cold, ultrasound and supercritical fluid extraction on the content of betalains, antioxidant, anti-inflammatory and antihyperglycemic activities.

Based on the obtained experimental results, the following was found:The extraction with water resulted in the highest extraction yield, especially for ultrasound and Soxhlet extraction, whereas the cold extraction gave a bit lower yield;The extraction yields with different solvents presented in the following order: H_2_O > 50% EtOH > 50% MeOH > 30% MeOH;Regarding the efficiency of the extraction method, the results showed that the yield of the extract was higher when extraction was done under reflux (SE); this suggests that hot solvent systems under reflux state are more efficient for recovering antioxidant components, thus offering higher extract yields;The extraction yields with different methods presented in the following order: SE > UE > CE;The yield of SFE with carbon dioxide and both co-solvents, ethanol and isopropanol, was significantly lower. The highest value was obtained with isopropanol as the co-solvent, under the pressure of 300 bar and a temperature of 60 °C, but was still significantly lower than those obtained using the above-mentioned extraction methods. This can be a consequence of the difference in chemical characteristics and polarities of various antioxidant compounds and particular solvents;The achieved total phenolic content in beetroot extracts ranged from 76.05 mg GAE/g to 226.80 mg GAE/g. The highest content of total phenolics was found in an extract prepared using SE 50% EtOH, but no significant differences were observed compared to the UE 30% EtOH and SE H_2_O;In SFE samples, phenolic compounds were not determined. Precisely, carbon dioxide is a non-polar molecule, whereas the targeted compounds (e.g., polyphenols and betalains) are mostly polar;Protocatechuic acid was the most abundant of the eight phenolic compounds detected in the extracts;The concentrations of total phenols ranged from 12.09 mg/g (UE 30% MeOH) to 18.60 mg/g (SE 50% EtOH);The highest anti-inflammatory activity (up to 42.86%) was observed for CE 50% MeOH extract at a concentration of 10 mg/mL;

The high radical scavenging activity of the SFE extracts could be a consequence of the presence of non-phenolic compounds. The experimental results were further analysed using the chemometrics approach to find the “greenest” method for the further possible application of beetroot extraction in the food and/or pharmaceutical industry. According to the standard score, the best extraction method was SE 50% EtOH, with an SS value of 0.674. Furthermore, the ANN model was used to predict antioxidant values which were very close to the target values of the antioxidant assays in terms of the R^2^ value.

## Figures and Tables

**Figure 1 antioxidants-11-00805-f001:**
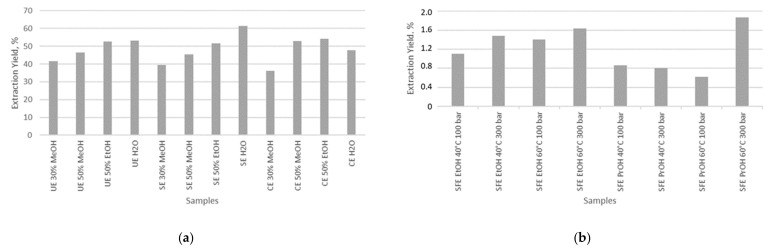
Extraction yield of beetroot extracts obtained by (**a**) ultrasound (UE), Soxhlet (SE), and cold extraction (CE) techniques, and (**b**) supercritical fluid extraction (SFE).

**Figure 2 antioxidants-11-00805-f002:**
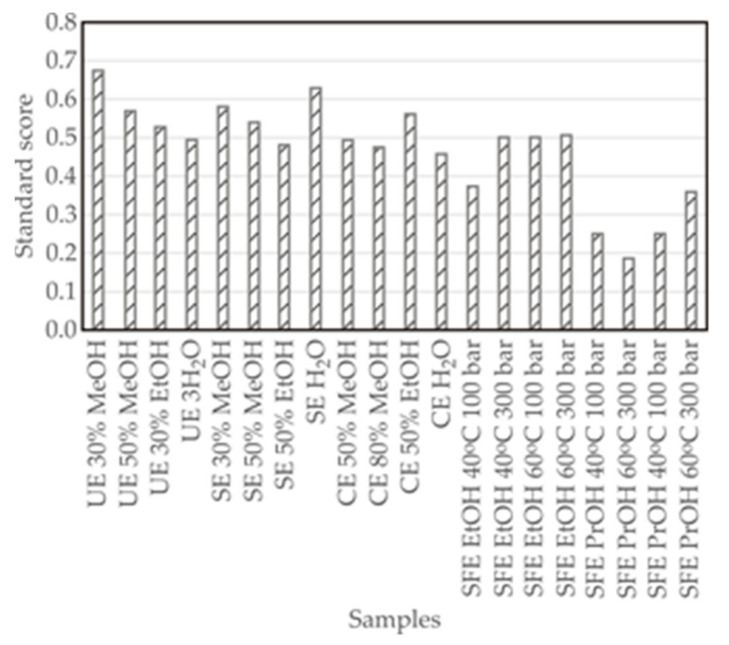
Standard score of extraction techniques: ultrasound (UE), Soxhlet (SE), cold extraction (CE) techniques and supercritical fluid extraction (SFE) based on total phytochemicals, antioxidant and pharmacological activities.

**Figure 3 antioxidants-11-00805-f003:**
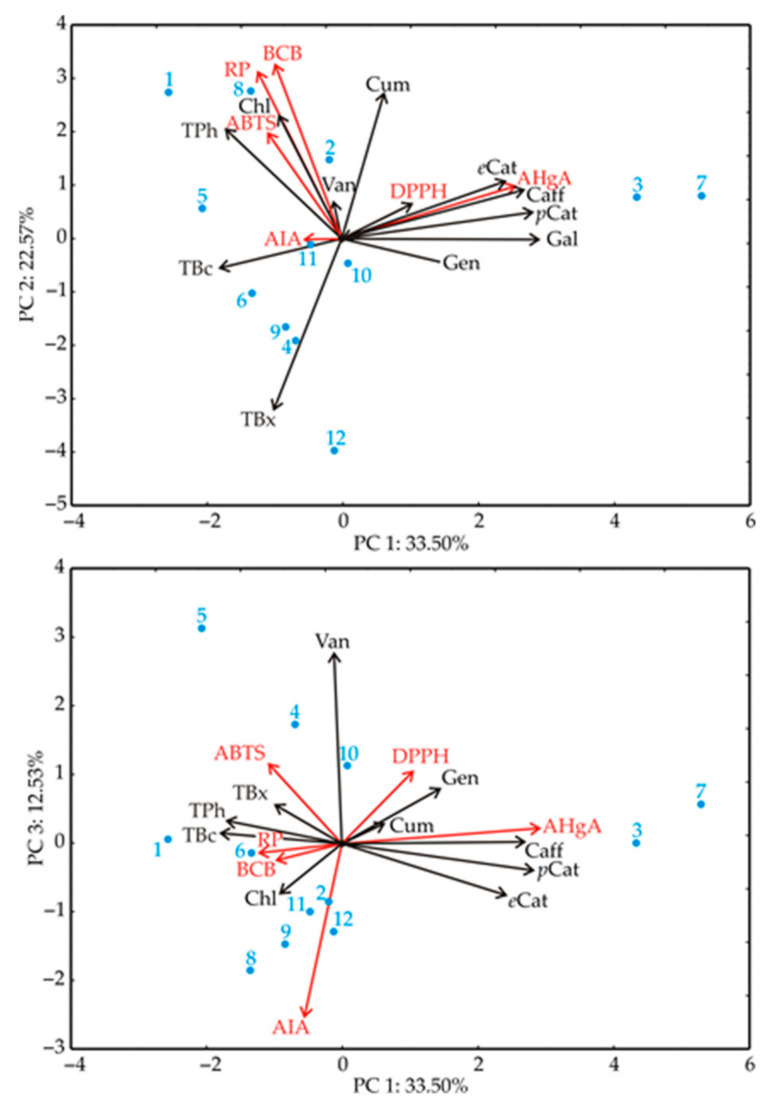
PCA ordination of variables based on correlations of phytochemicals, antioxidant and pharmacological activities.

**Figure 4 antioxidants-11-00805-f004:**
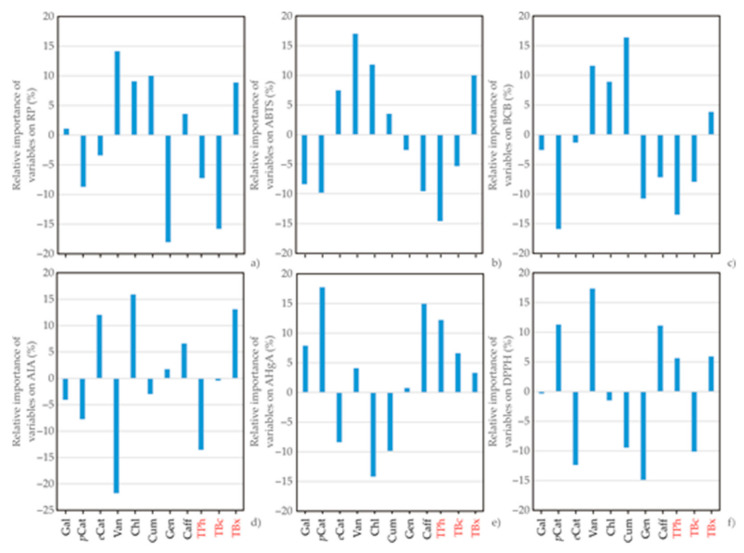
The relative importance of Gal, *p*Cat, *e*Cat, Van, Chl, Cum, Gen and Caff, and also TPh, TBc and TBx on (**a**) RP, (**b**) ABTS, (**c**) BCB, (**d**) AIA, (**e**) AHgA and (**f**) DPPH, determined using the Yoon interpretation method.

**Table 1 antioxidants-11-00805-t001:** Total phenolic and betalain compound contents of beetroot extracts.

Sample	TPh	TBc	TBx
UE 30% MeOH	135.80	2.67 ± 0.09 ^a^	2.59 ± 0.08 ^a^
UE 50% MeOH	95.44	2.73 ± 0.14 ^ab^	2.63 ± 0.15 ^a^
UE 30% EtOH	222.92	2.52 ± 0.12 ^a^	2.49 ± 0.13 ^a^
UE H_2_O	157.04	2.78 ± 0.12 ^abc^	3.33 ± 0.13 ^c^
SE 30% MeOH	196.77	4.17 ± 0.17 ^f^	3.41 ± 0.13 ^cd^
SE 50% MeOH	198.43	3.77 ± 0.24 ^ef^	3.37 ± 0.21 ^c^
SE 50% EtOH	226.80	3.15 ± 0.17 ^bcd^	2.67 ± 0.15 ^ab^
SE H_2_O	221.57	3.80 ± 0.15 ^ef^	2.38 ± 0.10 ^a^
CE 50% MeOH	76.05	3.63 ± 0.18 ^e^	3.29 ± 0.16 ^c^
CE 80% MeOH	182.74	3.17 ± 0.20 ^cd^	2.66 ± 0.16 ^ab^
CE 50% EtOH	181.02	3.12 ± 0.15 ^bc^	3.03 ± 0.14 ^bc^
CE H_2_O	138.51	3.56 ± 0.13 ^de^	3.77 ± 0.14 ^d^
SFE EtOH 40°C 100 bar	nd	nd	nd
SFE EtOH 40°C 300 bar	nd	nd	nd
SFE EtOH 60°C 100 bar	nd	nd	nd
SFE EtOH 60°C 300 bar	nd	nd	nd
SFE PrOH 40°C 100 bar	nd	nd	nd
SFE PrOH 40°C 300 bar	nd	nd	nd
SFE PrOH 60°C 100 bar	nd	nd	nd
SFE PrOH 60°C 300 bar	nd	nd	nd

Results are expressed as mean ± standard deviation (*n* = 3). nd—not detected. Values in the row with different superscripts are significantly different at *p* ≤ 0.05, according to Tukey’s HSD test. TPh—Total phenolics (mg GAE/g); TBc—Total betacyanins (mg BE/100 g); TBx—Total betaxanthins (mg VE/100 g).

**Table 2 antioxidants-11-00805-t002:** HPLC analysis of individual phenolic compounds in beetroot extracts.

Sample	Gallic Acid	Protocatechuic Acid	Epicatechin	Vanillic Acid	Chlorogenic Acid	Coumaric Acid	Gentisic Acid	Caffeic Acid	Total Content
**UE 30% MeOH**	1.00 ± 0.00 ^c^	10.09 ± 0.00 ^a^	0.13 ± 0.00 ^b^	0.50 ± 0.00 ^h^	0.15 ± 0.00 ^e^	0.03 ± 0.00 ^c^	0.18 ± 0.00 ^b^	0.01 ± 0.00 ^a^	12.09 ± 0.00 ^a^
**UE 50% MeOH**	1.20 ± 0.00 ^g^	11.51 ± 0.02 ^f^	1.19 ± 0.00 ^i^	0.28 ± 0.00 ^d^	0.09 ± 0.00 ^c^	0.02 ± 0.00 ^b^	0.20 ± 0.00 ^b^	0.01 ± 0.00 ^a^	14.50 ± 0.02 ^h^
**UE 30% EtOH**	1.72 ± 0.01 ^h^	13.89 ± 0.00 ^i^	1.64 ± 0.00 ^l^	0.44 ± 0.00 ^g^	0.17 ± 0.00 ^f^	0.02 ± 0.00 ^b^	0.30 ± 0.02 ^d^	0.06 ± 0.00 ^c^	18.24 ± 0.03 ^j^
**UE H_2_O**	1.12 ± 0.00 ^e^	10.96 ± 0.00 ^c^	0.35 ± 0.00 ^d^	0.50 ± 0.00 ^h^	0.04 ± 0.00 ^b^	0.01 ± 0.00 ^a^	0.19 ± 0.00 ^b^	0.01 ± 0.00 ^a^	13.18 ± 0.00 ^c^
**SE 30% MeOH**	0.97 ± 0.00 ^b^	10.80 ± 0.00 ^b^	0.20 ± 0.00 ^c^	0.68 ± 0.00 ^j^	0.11 ± 0.00 ^d^	0.04 ± 0.00 ^d^	0.20 ± 0.01 ^b^	0.02 ± 0.00 ^b^	13.02 ± 0.01 ^b^
**SE 50% MeOH**	1.09 ± 0.00 ^d^	11.49 ± 0.01 ^ef^	0.09 ± 0.00 ^a^	0.35 ± 0.00 ^e^	0.15 ± 0.01 ^e^	0.03 ± 0.00 ^c^	0.20 ± 0.01 ^b^	0.02 ± 0.00 ^b^	13.42 ± 0.03 ^d^
**SE 50% EtOH**	1.94 ± 0.00 ^i^	14.51 ± 0.01 ^j^	1.47 ± 0.00 ^k^	0.37 ± 0.00 ^f^	0.01 ± 0.00 ^a^	0.05 ± 0.00 ^e^	0.18 ± 0.00 ^b^	0.07 ± 0.00 ^d^	18.60 ± 0.01 ^k^
**SE H_2_O**	0.93 ± 0.00 ^a^	12.12 ± 0.01 ^h^	0.79 ± 0.00 ^h^	0.11 ± 0.00 ^a^	0.21 ± 0.01 ^g^	0.04 ± 0.00 ^d^	0.14 ± 0.00 ^a^	0.02 ± 0.00 ^b^	14.36 ± 0.02 ^g^
**CE 50% MeOH**	1.21 ± 0.00 ^g^	11.70 ± 0.00 ^g^	0.49 ± 0.00 ^f^	0.22 ± 0.00 ^c^	0.08 ± 0.00 ^c^	0.02 ± 0.00 ^b^	0.19 ± 0.00 ^b^	0.02 ± 0.00 ^b^	13.93 ± 0.00 ^e^
**CE 80% MeOH**	1.21 ± 0.00 ^g^	11.47 ± 0.00 ^e^	0.51 ± 0.00 ^g^	0.51 ± 0.00 ^i^	0.08 ± 0.00 ^c^	0.02 ± 0.00 ^b^	0.23 ± 0.01 ^c^	0.01 ± 0.00 ^a^	14.04 ± 0.01 ^f^
**CE 50% EtOH**	0.99 ± 0.00 ^c^	11.71 ± 0.02 ^g^	1.22 ± 0.00 ^j^	0.35 ± 0.00 ^e^	0.11 ± 0.00 ^d^	0.02 ± 0.00 ^b^	0.18 ± 0.00 ^b^	0.02 ± 0.00 ^b^	14.60 ± 0.02 ^i^
**CE H_2_O**	1.16 ± 0.01 ^f^	11.23 ± 0.00 ^d^	0.41 ± 0.00 ^e^	0.13 ± 0.00 ^b^	0.04 ± 0.00 ^b^	0.01 ± 0.00 ^a^	0.19 ± 0.00 ^b^	0.02 ± 0.00 ^b^	13.19 ± 0.01 ^c^

Results are expressed as mean ± standard deviation (*n* = 3). Values in the row with different superscripts are significantly different at *p* ≤ 0.05, according to Tukey’s HSD test. The contents of phenolic compounds are expressed as mg/g.

**Table 3 antioxidants-11-00805-t003:** Antioxidant and pharmacological activities of beetroot extracts.

Sample	DPPH	ABTS	RP	BCB	AIA	AHgA
UE 30% MeOH	10.26 ± 1.15 ^cd^	2774.14 ± 95.91 ^b^	147.03 ± 7.37 ^bcd^	131.61 ± 0.11 ^cdef^	17.62 ± 0.76 ^bc^	67.99 ± 3.82 ^fg^
UE 50% MeOH	11.36 ± 0.53 ^d^	3809.27 ± 44.89 ^hi^	143.25 ± 7.37 ^bcd^	133.21 ± 0.72 ^cdefg^	37.08 ± 1.88 ^ef^	56.60 ± 3.40 ^def^
UE 30% EtOH	7.33 ± 0.69 ^ab^	3798.23 ± 14.20 ^hi^	191.07 ± 14.35 ^efg^	161.15 ± 8.47 ^hi^	33.64 ± 1.93 ^e^	46.40 ± 6.52 ^abcd^
UE H_2_O	11.40 ± 0.38 ^d^	3964.30 ± 23.23 ^j^	133.95 ± 14.07 ^bc^	123.98 ± 15.24 ^cde^	22.15 ± 1.43 ^cd^	38.82 ± 9.13 ^ab^
SE 30% MeOH	10.53 ± 1.16 ^cd^	4566.45 ± 55.75 ^l^	177.23 ± 20.97 ^def^	148.79 ± 12.70 ^fgh^	12.92 ± 3.74 ^b^	37.66 ± 1.01 ^ab^
SE 50% MeOH	7.22 ± 0.70 ^ab^	3560.76 ± 62.35 ^ef^	138.89 ± 4.20 ^bcd^	116.38 ± 6.74 ^bcd^	35.49 ± 2.66 ^e^	42.81 ± 6.30 ^abc^
SE 50% EtOH	7.26 ± 0.37 ^ab^	4258.90 ± 36.56 ^k^	278.00 ± 22.35 ^h^	174.29 ± 13.64 ^ij^	37.64 ± 4.17 ^ef^	46.73 ± 1.68 ^abcd^
SE H_2_O	11.13 ± 0.30 ^d^	3520.42 ± 47.42 ^e^	209.26 ± 7.50 ^fg^	164.41 ± 12.35 ^hi^	36.53 ± 1.91 ^e^	40.42 ± 7.53 ^ab^
CE 50% MeOH	5.79 ± 1.92 ^a^	3752.45 ± 67.91 ^gh^	149.96 ± 26.61 ^bcde^	137.28 ± 4.71 ^defg^	42.86 ± 1.67 ^f^	34.19 ± 3.33 ^a^
CE 80% MeOH	6.11 ± 0.35 ^a^	3652.92 ± 54.49 ^fg^	138.12 ± 5.19 ^bcd^	128.01 ± 3.65 ^cdef^	21.41 ± 0.80 ^cd^	49.61 ± 2.73 ^bcde^
CE 50% EtOH	6.03 ± 1.09 ^a^	4388.60 ± 66.94 ^k^	147.64 ± 2.84 ^bcd^	130.93 ± 8.45 ^cdef^	42.68 ± 1.22 ^f^	46.87 ± 2.91 ^abcd^
CE H_2_O	8.58 ± 1.13 ^bc^	1559.99 ± 23.02 ^a^	108.76 ± 2.99 ^ab^	113.26 ± 1.53 ^bc^	35.42 ± 0.32 ^e^	44.20 ± 1.29 ^abcd^
SFE EtOH 40 °C 100 bar	19.87 ± 0.49 ^e^	4527.87 ± 19.39 ^l^	209.53 ± 22.82 ^fg^	155.60 ± 0.51 ^ghi^	17.38 ± 1.76 ^bc^	35.60 ± 5.45 ^a^
SFE EtOH 40 °C 300 bar	19.86 ± 0.45 ^e^	4889.60 ± 15.79 ^m^	219.21 ± 5.85 ^fg^	142.15 ± 7.85 ^efgh^	23.59 ± 3.37 ^d^	88.08 ± 5.55 ^h^
SFE EtOH 60 °C 100 bar	20.03 ± 0.59 ^e^	5362.63 ± 15.27 ^n^	221.54 ± 27.65 ^g^	156.57 ± 6.61 ^ghi^	24.51 ± 2.53 ^d^	73.25 ± 7.58 ^g^
SFE EtOH 60 °C 300 bar	11.51 ± 0.64 ^d^	5786.35 ± 20.53 ^o^	296.78 ± 15.27 ^h^	193.07 ± 10.02 ^j^	23.62 ± 1.41 ^d^	67.79 ± 3.17 ^fg^
SFE PrOH 40 °C 100 bar	20.07 ± 0.58 ^e^	3307.98 ± 44.58 ^d^	108.50 ± 10.50 ^ab^	99.54 ± 9.14 ^b^	1.11 ± 0.74 ^a^	61.98 ± 1.11 ^efg^
SFE PrOH 40 °C 300 bar	20.39 ± 0.61 ^e^	1654.50 ± 22.97 ^a^	90.56 ± 5.07 ^a^	67.06 ± 0.71 ^a^	2.08 ± 0.91 ^a^	67.98 ± 2.72 ^fg^
SFE PrOH 60 °C 100 bar	20.13 ± 0.67 ^e^	2973.34 ± 62.09 ^c^	109.58 ± 14.31 ^ab^	113.84 ± 9.82 ^bcd^	4.07 ± 0.94 ^a^	55.72 ± 4.21 ^cdef^
SFE PrOH 60 °C 300 bar	19.82 ± 0.50 ^e^	3886.27 ± 15.60 ^ij^	163.31 ± 17.49 ^cde^	145.94 ± 2.09 ^efgh^	6.891.63 ^a^	66.42 ± 3.72 ^fg^

Results are expressed as mean ± standard deviation (*n* = 3). Values in the row with different superscripts are significantly different at *p* ≤ 0.05, according to Tukey’s HSD test. DPPH—DPPH free radical scavenging assay (μmol TE/100 g); ABTS—ABTS free radical scavenging assay (μmol TE/100 g); RP—reducing power (μmol TE/100 g); BCB—β-carotene bleaching assay (μmol TE/100 g); AIA—anti-inflammatory activity (% of inhibition); AHgA—antihyperglycemic activity (% of inhibition).

## Data Availability

Data is contained within the article.

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
