# Peer review of "Green Techniques for Preparation of Red Beetroot Extracts with Enhanced Biological Potential"

_antioxidants, 2022, doi:10.3390/antiox11050805_

Round 1
Reviewer 1 Report
This paper presented extraction techniques for preparation of red beetroot extracts with enhanced biological potential but some revisions are required for better quality of your paper including mimic errors.
- Please avoid a phrase `in order to ---’.
- Please correct H2O and CO2 in your text.
- Your test sample (beet root) may be powder-type. Extraction yield can be different according to particle size. How much was particle size of your samples
- Please define `extraction yield’ in `Materials and Methods'.
- You need discuss on results of AIA test including mechanism of anti-Inflammatory effect. I think that inflammation may be very complicated by various factors. Can you say that this extract has anti-inflammatory effect only due to inhibition of AIA?
- In Table 3, insert units of each column (each antioxidant assay). Any SFE is not effective for extraction of phenolics and betalain (Table 1). Then, how have SFE extracts antioxidant effects?
- In line 147, 1% formic acid in d-water?
Author Response
We appreciate your and the reviewers' time and effort reviewing our Manuscript. We are submitting the corrected version of the Manuscript entitled “Green techniques for preparation of red beetroot extracts with enhanced biological potential” to be reviewed and potentially published in the Antioxidants journal.
We believe that the remarks were followed precisely and incorporated improvements (please see changes highlighted with yellow color). Therefore, we are firmly convinced that the revised Manuscript meets the journal publication requirements.
Comments:
This paper presented extraction techniques for preparation of red beetroot extracts with enhanced biological potential but some revisions are required for better quality of your paper including mimic errors.
- Please avoid a phrase `in order to ---’.
Response: “In order to” has been changed with “to” (please check track changes in the Manuscript).
- Please correct H2O and CO2 in your text.
Response: It has been corrected (please check track changes in the Manuscript).
- Your test sample (beet root) may be powder-type. Extraction yield can be different according to particle size. How much was particle size of your samples
Response: The material was ground (d ≈ 0.5 mm) before the extraction, but it was not powder – powder would stick to the walls of the extractor due to the high content of saccharides.
- Please define `extraction yield’ in `Materials and Methods'.
Response: It has been defined in subsection 2.3. Preparation of beetroot extracts (please check track changes in the Manuscript).
- You need discuss on results of AIA test including mechanism of anti-Inflammatory effect. I think that inflammation may be very complicated by various factors. Can you say that this extract has anti-inflammatory effect only due to inhibition of AIA?
Response: The mechanism of AIA was discussed, as requested (please check track changes in the Manuscript). Based on this, it was concluded that the anti-inflammatory effect of this extract is the possible biological outcome.
- In Table 3, insert units of each column (each antioxidant assay). Any SFE is not effective for extraction of phenolics and betalain (Table 1). Then, how have SFE extracts antioxidant effects?
Response: Units have been added below Table 3, as requested (please check track changes in the Manuscript). Antioxidant activity can be expressed through various mechanisms. SFE extracts contain other phytochemicals, which have the polarity of supercritical fluids used for extraction and show antioxidant activity as well, depending on the chemical structures of the components.
Antioxidant activity can be expressed through various mechanisms. SFE extracts contain other phytochemicals, which have lower polarity due to the fact that supercritical CO2 in a presence of a polar cosolvent has been used for extraction. Beetroot is rich in carbohydrates, fat, protein, micronutrients and several functional constituents having substantial health-promoting properties. Among less polar and non-polar compounds; beetroot contain the considerable amount of carotenoids and both essential and non-essential amino acids such as tryptophan, isoleucine, leucine, lysine, threonine, methionine, phenylalanine, tyrosine, valine, cystine, arginine, histidine, alanine, glutamic acid, glycine, proline, aspartic acid and serine [49]. These compounds act as anticarcinogens and immunoenhancers, involve pro-vitamin A activity and poses antioxidant ability. Supercritical fluid extraction was used to extract free amino acids from sugar beet and sugar cane molasses [50], which indicates the presence of these compounds in supercritical extracts.
- In line 147, 1% formic acid in d-water?
Response: Yes, it has been specified (please check track changes in the Manuscript).
Reviewer 2 Report
Manuscript ID: antioxidants-1681054
Type of manuscript: Article
Title: Green techniques for preparation of red beetroot extracts with enhanced biological potential
Authors: Dragana Borjan , Vanja Šeregelj , Darija Cör Andrejč , Lato Pezo , Vesna Tumbas Šaponjac , Željko Knez , Jelena Vulić , Maša Knez Marevci *
Review of the manuscript
Comments
The manuscript presented for revision is very interesting. The obtained results are discussed with the works published in recent years. This work is well organized and scientifically sound. However, the authors did not avoid some minor mistakes that we all make while preparing the publications. I only have some minor comments:
- Was only one batch of lyophilized beetroot material used in the research?
- Conclusions (page 12 line 455)
Please expand the techniques abbreviations: SE, CE, UE and SFE in the sentence: "This study aimed to investigate the influence of isolation techniques such as SE, CE, UE and SFE on the content of betalains, antioxidant, anti-inflammatory and antihyperglycemic activities. "
Author Response
Reviewer: 2
We appreciate your time and effort reviewing our Manuscript. We are submitting the corrected version of the Manuscript entitled “Green techniques for preparation of red beetroot extracts with enhanced biological potential” to be reviewed and potentially published in the Antioxidants journal.
Your suggestions have been addressed - throughout the Manuscript, the corrections have been marked with blue.
Comments:
The manuscript presented for revision is very interesting. The obtained results are discussed with the works published in recent years. This work is well organized and scientifically sound. However, the authors did not avoid some minor mistakes that we all make while preparing the publications. I only have some minor comments:
- Was only one batch of lyophilized beetroot material used in the research?
Response: Yes, the used material for all extractions has been from the same batch. It has been defined in the Materials and Methods section (please check track changes in the Manuscript).
- Conclusions (page 12 line 455)
Please expand the techniques abbreviations: SE, CE, UE and SFE in the sentence: "This study aimed to investigate the influence of isolation techniques such as SE, CE, UE and SFE on the content of betalains, antioxidant, anti-inflammatory and antihyperglycemic activities. "
Response: It has been corrected as suggested (please check track changes in the Manuscript).
Round 2
Reviewer 1 Report
All points recommended by me were responded or revised very well.